# Aggregation-regulated room-temperature phosphorescence materials with multi-mode emission, adjustable excitation-dependence and visible-light excitation

Jingxuan You[1,2], Xin Zhang[1], Qinying Nan[1,2], Kunfeng Jin[1], Jinming Zhang ®[1]✉, Yirong Wang[1,2], Chunchun Yin[1], Zhiyong Yang ®[3]✉ & Jun Zhang ®[1,2]

Constructing room-temperature phosphorescent materials with multiple emission and special excitation modes is fascinating and challenging for practical applications. Herein, we demonstrate a facile and general strategy to obtain ecofriendly ultralong phosphorescent materials with multi-mode emission, adjustable excitation-dependence, and visible-light excitation using a single organic component, cellulose trimellitate. Based on the regulation of the aggregation state of anionic cellulose trimellitates, such as CBtCOONa, three types of phosphorescent materials with different emission modes are fabricated, including blue, green and color-tunable phosphorescent materials with a strong excitation-dependence. The separated molecularly-dispersed CBtCOONa exhibits blue phosphorescence while the aggregated CBtCOONa emits green phosphorescence; and the CBtCOONa with a coexistence state of single molecular chains and aggregates exhibits color-tunable phosphorescence depending on the excitation wavelength. Moreover, aggregated cellulose trimellitates demonstrate unique visible-light excitation phosphorescence, which emits green or yellow phosphorescence after turning off the visible light. The aggregation-regulated phenomenon provides a simple principle for designing the proof-of-concept and on-demand phosphorescent materials by using a single organic component. Owing to their excellent processability and environmental friendliness, the aforementioned cellulose-based phosphorescent materials are demonstrated as advanced phosphorescence inks to prepare various disposable complex anticounterfeiting patterns and information codes.

Organic room-temperature phosphorescence (RTP) materials have unique photophysical properties, rich excited-state features, good biocompatibility, high flexibility, low cost, and excellent structural designability[1,2]. In addition, the nature of phosphorescence, with respect to the radiative transition from the excited triplet states ($T_n$) to the ground states ($S_0$), endows RTP materials with long emission lifetime and large Stokes shift[3,4]. Therefore, organic RTP materials have recently attracted tremendous interest, and indicate attractive

[1]CAS Key Laboratory of Engineering Plastics, Institute of Chemistry, Chinese Academy of Sciences (CAS), Beijing 100190, China. [2]University of Chinese Academy of Sciences, Beijing 100049, China. [3]School of Chemistry, Sun Yat-sen University, Guangzhou 510275, China. ✉e-mail: zhjm@iccas.ac.cn; yangzhy29@mail.sysu.edu.cn

application prospects in various fields, such as bioimaging[5], optical recording[6], information storage[7], anticounterfeiting system[8], and organic light-emitting diodes[9–11].

Multi-color and stimuli-responsive RTP materials can collect more information and provide advanced utilization. Therefore, they are highly desired in the phosphorescence field[12–16]. An et al. designed and synthesized a series of carbazole derivatives to achieve multicolor phosphorescence[12]. Chen et al. introduced different brominated aromatic groups into polyacrylamide side chains to obtain RTP materials with phosphorescence colors from green to orange[17]. Lei et al. prepared multicolor RTP materials by changing the chemical structure of the donor and acceptor[18]. Wang et al. modified the side groups on the aromatic structure to obtain multicolor RTP materials[19]. Tan et al. introduced different fluorescence groups into the biopolymer sodium alginate (SA) via an amidation reaction to achieve color-tunable RTP materials[20]. Furthermore, they doped different aromatic carboxylates into SA to obtain colorful and time-responsive afterglow materials with adjustable colors from blue to red[21]. Zhang et al. combined polyvinyl alcohol (PVA) and heterocyclic aromatic molecules to form large-area multi-color phosphorescent films[22,23]. So far, the preparation of these multicolor RTP materials typically involves a laborious and time-consuming synthetic procedure, or entails a coordination of several chromophores with various chemical structures[24,25]. In addition, there have been many attempts to fabricate stimuli-responsive phosphorescence materials, including force-responsive[26–28], pH-responsive[29–31], light-responsive[32–34], temperature-responsive[35,36], humidity-responsive[37], and redox-responsive materials[38]. For instance, Chi et al. reported an aggregation-induced emission (AIE) luminogen of 2-([1,1':3',1''-terphenyl]-5'-yl)-4,4,5,5-tetramethyl-1,3,2-dioxaborolane exhibiting fluorescence-phosphorescence dual emission under mechanical stimulation[39,40]. Gong et al. prepared pH-responsive RTP materials by mixing cururbit[7]uril (CB[7]) with a 6-bromoisoquinoline derivative[41]. The switching RTP emission of the molecular shuttle could be detected with the naked eye by altering the pH. Yuan and Tan produced a series of excitation-dependent RTP materials based on a clustering-triggered emission mechanism[20]. Cai et al. found that the poly(styrene sulfonic acid) sodium exhibited excitation-dependent phosphorescence at 77 K based on the same mechanism[42]. The excitation-dependent phosphorescence materials exhibit varied phosphorescence colors under different excitation wavelengths. Therefore, they are promising candidates for high-security information storage and anticounterfeiting applications. Furthermore, the detection method for excitation-dependent RTP materials is simple and easy to use. However, the responsive behavior of such excitation-dependent RTP materials is nonadjustable. When they are used to prepare complicated phosphorescence patterns, other types of RTP materials are necessary, such as RTP materials without excitation wavelength dependence. Remarkably, RTP materials are generally used as light and thin materials, such as patterns, labels, coatings, and films, which are difficult to be recycled. Hence, it is desirable for these materials to be completely degradable. However, it is difficult to achieve complete biodegradation in case of synthetic polymers. Hence, it is fascinating and pragmatic to prepare RTP materials with multiple emission modes, special excitation modes, and excellent biodegradability by developing a facile preparation strategy using sustainable components.

Herein, we demonstrated a principle for regulating phosphorescence. Ecofriendly ultralong RTP materials with multi-mode emission, adjustable excitation-dependence, and visible-light excitation were fabricated using a single biopolymer component and regulating its aggregation state.

## Results

### Multimode RTP materials

Natural cellulose with a strong hydrogen-bonding network is an ideal substrate for constructing RTP materials. Anionic phenylcarboxylate substituents were added into the cellulose chain via chemical immobilization to obtain cellulose derivatives with ultralong RTP, such as cellulose trimellitate (Fig. S1a). In the $^1$H-NMR spectrum of cellulose trimellitate (CBtCOOH), the peaks observed in the 7.5–8.5 ppm range are assigned to the protons on the benzene ring while the peaks observed at 2.7–5.5 ppm region correspond to the protons of the cellulose backbone (Fig. S1b). In the FTIR spectrum of CBtCOOH, the peak at 1703 cm$^{-1}$ corresponds to the C = O stretching vibration peak (Fig. S1c). These results prove that the cellulose derivative, CBtCOOH, was successfully synthesized. Via controlling the reaction time and the molar ratio of trimellitic anhydride and anhydroglucose unit (AGU), a series of CBtCOOH with different degree of substitution (DS) from 0.33 to 1.12 were synthesized (Table S1). Subsequently, after a neutralization reaction between CBtCOOH and NaHCO$_3$, sodium cellulose trimellitate (CBtCOONa) was obtained. The quantum yield and RTP lifetime of CBtCOONa (DS = 0.54) were higher than those of other samples, thus it was used as the raw materials. The amorphous CBtCOONa powder exhibited blue fluorescence and green phosphorescence under irradiation with a 365 nm lamp and with the lamp off, respectively (Fig. S1d and S2).

Surprisingly, the CBtCOONa aqueous solutions of different concentrations exhibited different phosphorescence phenomena at 77 K (Fig. S3). At 0.2 mg/mL concentration, the CBtCOONa aqueous solution exhibited blue phosphorescence at 310 nm excitation while no phosphorescence at 365 nm excitation. Conversely, at 60 mg/mL concentration, the CBtCOONa aqueous solution exhibited blue phosphorescence at 310 nm excitation and green phosphorescence at 365 nm excitation. After adding a quantity of 100 mM CaCl$_2$ into the 0.2 mg/mL CBtCOONa aqueous solution, the obtained solution emitted both blue phosphorescence at 310 nm excitation and green phosphorescence at 365 nm excitation. We speculated that this phenomenon was related to the solution state of CBtCOONa. At 0.2 mg/mL concentration, the CBtCOONa aqueous solution was a dilute solution, in which the polymer chains were molecularly dispersed. The independent sodium trimellitate group exhibited blue phosphorescence. When the concentration of CBtCOONa aqueous solution increased to 60 mg/mL, the polymer chains strongly entangled with each other. Consequently, the sodium trimellitate groups formed aggregated structures, which emitted green phosphorescence. According to the plot of specific viscosity ($\eta_{sp}$) versus concentration, the overlap concentration (c*) and entanglement concentration (c$_e$) of the CBtCOONa aqueous solution are 0.2 wt% and 2.0 wt%, respectively (Fig. S4). Therefore, when the concentration was <0.2 wt%, the CBtCOONa aqueous solution (e.g., 0.2 mg/mL) was a molecularly-dispersed solution. When the concentration was >2.0 wt%, the CBtCOONa aqueous solution (e.g., 60.0 mg/mL) was in an entanglement state. These results confirm our above speculation. Furthermore, the appearance of green phosphorescence after adding the CaCl$_2$ solution into the 0.2 mg/mL CBtCOONa aqueous solution also proves the aforementioned mechanism, because Ca$^{2+}$ ions formed a chelate bond with COO$^-$ ions on the CBtCOONa chains, causing the aggregation of the polymer chains. Overall, the CBtCOONa aqueous solutions exhibited an aggregation-regulated phosphorescence phenomenon.

Inspired by the phosphorescence phenomenon of CBtCOONa solution at 77 K, we proposed to preserve the molecularly-dispersed state, the aggregate state and the coexistence state of single molecular chains and aggregates in the solid state. As a result, a series of RTP materials with different emission modes were obtained by controlling the ratio of the molecularly dispersed and aggregated states of CBtCOONa (Fig. 1 and S5). We first added a Na$_2$CO$_3$ solution into the CBtCOONa dilute solution to isolate CBtCOONa polymer chains using CO$_3^{2-}$ ions. Subsequently, a CaCl$_2$ solution was added into the aforementioned solution, resulting in immediate formation of insoluble CaCO$_3$. The molecularly-dispersed CBtCOONa chains were immobilized in CaCO$_3$ to prevent the formation of an aggregated state.

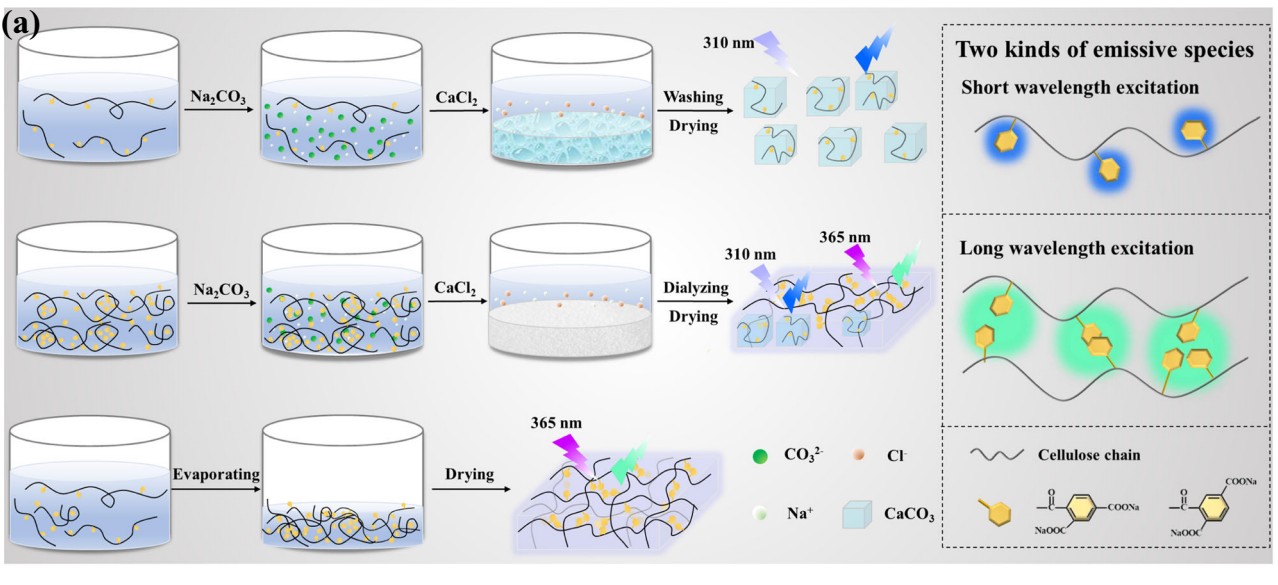

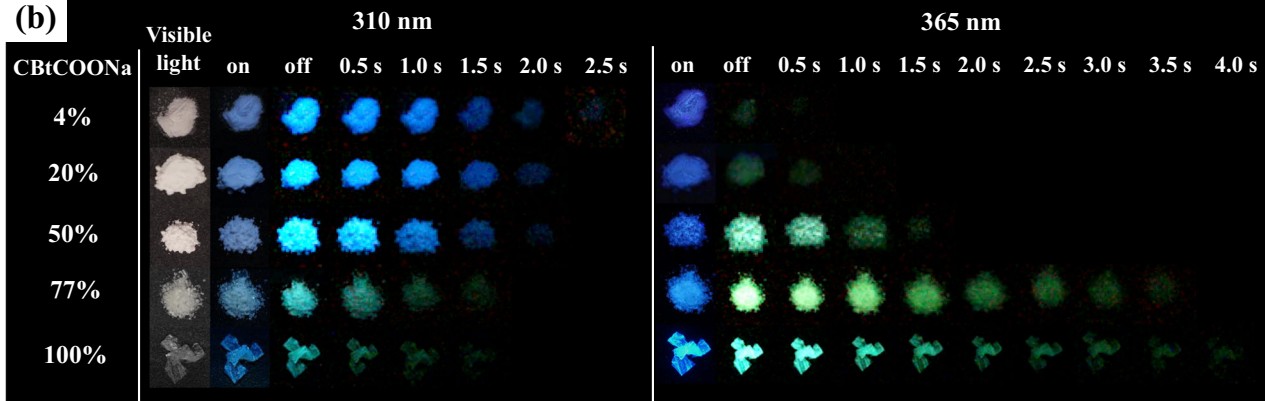

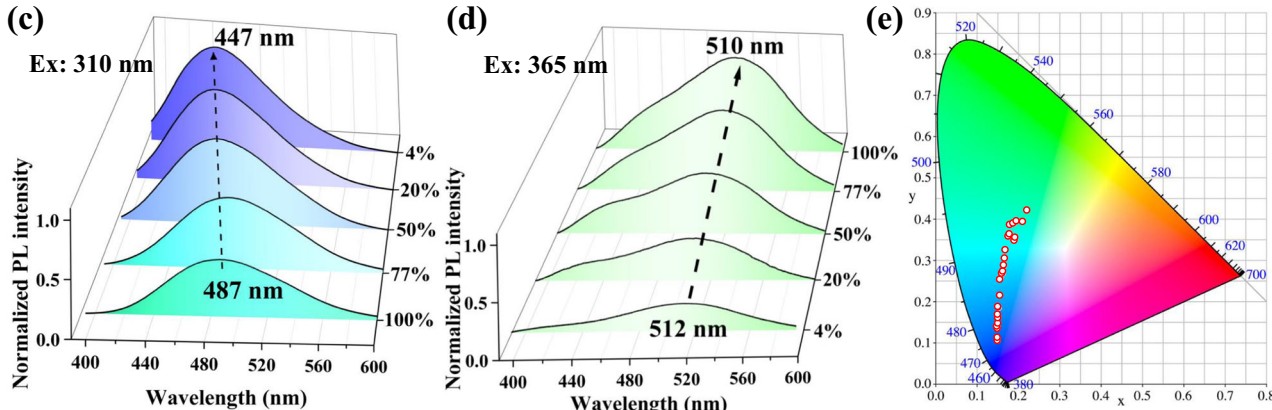

**Fig. 1 | Mechanism and properties of multi-mode RTP materials. a** Schematic illustration of sample preparation processes and microstructures; **b** Photographs of RTP materials with different CBtCOONa contents under irradiation with 310 and 365 nm lamps and with the lamps off; **c, d** Phosphorescence spectra of RTP materials with different CBtCOONa content at 310 nm and 365 nm excitations; **e** Commission Internationale d'Eclairage (CIE) coordinate diagram of RTP materials.

Consequently, the obtained solid material retained the phosphorescence property of the dilute CBtCOONa solution and emitted only blue phosphorescence (Fig. 1a, b). When $Na_2CO_3$ and $CaCl_2$ were sequentially added into the concentrated CBtCOONa solution, the obtained material had both molecularly-dispersed CBtCOONa chains and aggregated states. Therefore, the resultant material exhibited blue and green phosphorescence (Fig. 1a, b). When the solvent in the dilute CBtCOONa solution was slowly volatilized, the polymer chains gradually overlapped together, indicating an increasing entanglement degree. The final product was almost always the aggregated state of CBtCOONa, so it exhibited green phosphorescence. Thus, a series of RTP materials with different emission modes can be obtained by regulating the aggregation states of CBtCOONa.

The CBtCOONa/$CaCO_3$ powders with a CBtCOONa content of 4–20% prepared from dilute CBtCOONa solutions emitted bright blue phosphorescence at 310 nm (Fig. 1b and S5). The phosphorescence emission lasted for >2 s at room temperature after irradiation with a 310 nm ultraviolet lamp, as observed by the naked eye. The

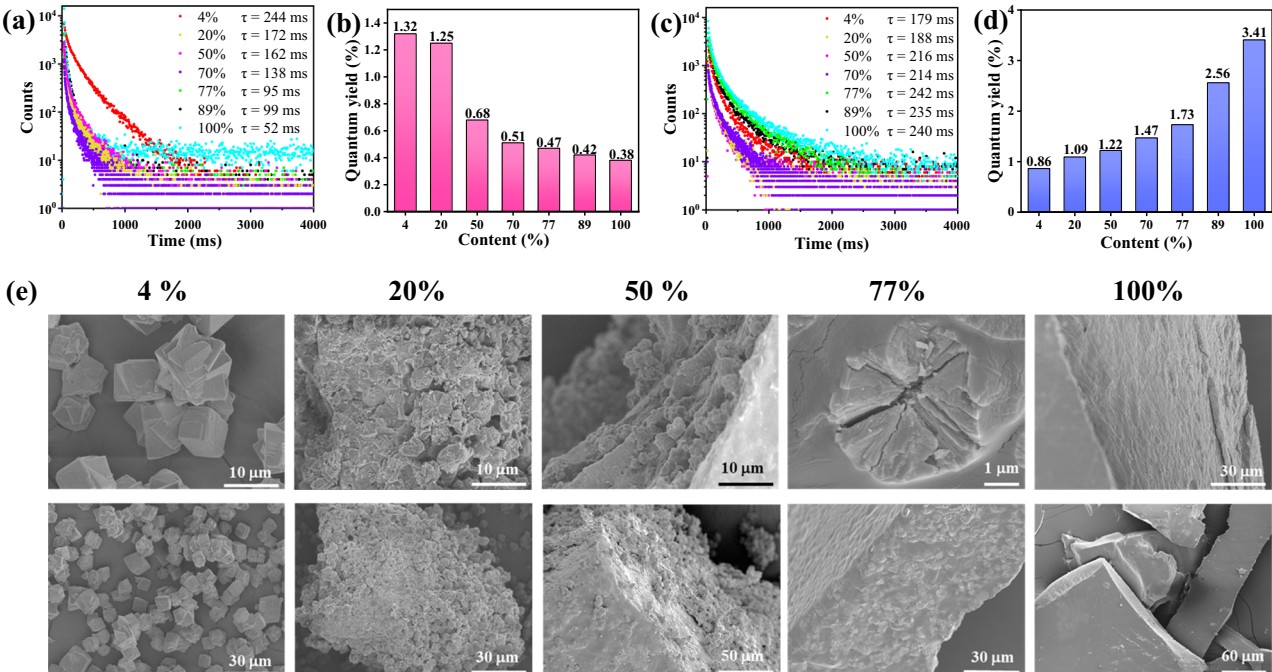

**Fig. 2 | RTP properties and microstructures of CBtCOONa/CaCO₃.**
**a** Phosphorescence lifetime of RTP materials with different CBtCOONa contents (Ex = 310 nm, detection wavelength = 450 nm); **b** Quantum yield of RTP materials with different CBtCOONa contents (Ex = 310 nm); **c** Phosphorescence lifetime of RTP materials with different CBtCOONa contents (Ex = 370 nm, detection wavelength = 500 nm); **d** Quantum yield of RTP materials with different CBtCOONa contents (Ex = 370 nm); **e** Scanning electron microscope (SEM) images of RTP materials with different CBtCOONa contents.

CBtCOONa/CaCO₃ powder with a CBtCOONa content of 50% prepared from the concentrated CBtCOONa solution exhibited the changeable RTP with a strong excitation dependence. It exhibited blue phosphorescence at 254–310 nm excitation and green phosphorescence at 340–365 nm excitation (Fig. 1b and S5). The persistent luminescence time of blue and green phosphorescence was >1.5 s. The pure CBtCOONa solid material had green phosphorescence with a weak excitation dependence at 310–365 nm excitation (Fig. 1b and S5). The persistent luminescence time was >3.5 s. Obviously, with the increase of the CBtCOONa content in CBtCOONa/CaCO₃ powder, the intensity and lifetime of blue phosphorescence gradually diminished, whereas the intensity and lifetime of green phosphorescence rapidly increased (Figs. 1b–d and 2a–d). Almost all the RTP materials had a phosphorescence lifetime of >100 ms, indicating an ultralong phosphorescence property (Fig. 2a–d). In the scanning electron microscope (SEM) images of the CBtCOONa/CaCO₃ powder with 4% CBtCOONa, cube-shaped CaCO₃ calcite crystals were observed; no organic matter was visible. In other words, the CBtCOONa chains were encapsulated in CaCO₃. When the content of CBtCOONa was increased to 20%, CaCO₃ crystals were adhered together by organic CBtCOONa. When the content of CBtCOONa was increased to 50–77%, CaCO₃ crystals were embedded in the lamellar structure formed by CBtCOONa, creating a coexistence state of single molecular chains and aggregates of CBtCOONa. Pure CBtCOONa has a dense structure at the surface and cross-section, indicating that CBtCOONa chains tightly stack and exist in the aggregated state. These results prove that various RTP materials with different emission modes were obtained via regulating the aggregation state of CBtCOONa (Fig. 1e and S5, and Table S2). The molecularly-dispersed CBtCOONa exhibited blue phosphorescence without an excitation-dependence; the aggregated CBtCOONa emitted green phosphorescence with a weak excitation-dependence. On the other hand, the CBtCOONa with a coexistence state of single molecular chains and aggregates exhibited color-tunable phosphorescence with a strong excitation-dependence.

## Phosphorescence mechanism

The solution state of CBtCOONa was examined to verify further the formation mechanism of the aforementioned RTP materials (Fig. 3a). Dynamic light scattering (DLS) shows that CBtCOONa chains were molecularly dispersed at a concentration of 0.2 mg/mL. When the concentration was increased to 60.0 mg/mL, CBtCOONa chains formed aggregates with different sizes. After adding 100 mM CaCl₂ into 0.2 mg/mL CBtCOONa aqueous solution, the hydrodynamic radius ($R_h$) became remarkably smaller. The formation of chelate bonds between the introduced Ca²⁺ ions and COO⁻ anions caused intra-chain and inter-chain aggregation of the CBtCOONa chains; thus, the volume of the CBtCOONa chains was compressed. The 0.2 mg/mL CBtCOONa aqueous solution emitted blue phosphorescence at 310 nm excitation and 77 K. Moreover, it had negligible phosphorescence at 370 nm excitation (Fig. 3b). The lifetime of blue phosphorescence was as high as 498 ms (Fig. 3c). The 60 mg/mL CBtCOONa aqueous solution emitted strong green phosphorescence at 370 nm excitation and 77 K, while had weak blue phosphorescence at 310 nm excitation (Fig. 3d). The lifetime of green phosphorescence was as high as 530 ms (Fig. 3e). Therefore, the blue phosphorescence and green phosphorescence originate from the molecularly dispersed CBtCOONa chains and the aggregates of CBtCOONa chains, respectively (Fig. 3f).

We prepared sodium trimellitate (BtCOONa) to illustrate the phosphorescence mechanism (Fig. S6). The dilute solution of BtCOONa (0.1 mg/mL) exhibited the same phosphorescence property as that of CBtCOONa dilute solution at 77 K. It emitted blue phosphorescence at 310 nm excitation while negligible phosphorescence at 365 nm excitation (Fig. S7). The concentrated solution of BtCOONa (400 mg/mL) exhibited the same phosphorescence performance as that of CBtCOONa concentrated solution at 77 K. It emitted green phosphorescence at 365 nm excitation and blue phosphorescence emission at 310 nm excitation (Fig. S7). With the increase of temperature, the phosphorescence lifetime of BtCOONa decreased gradually (Fig. S8). These phenomena illustrate that blue phosphorescence

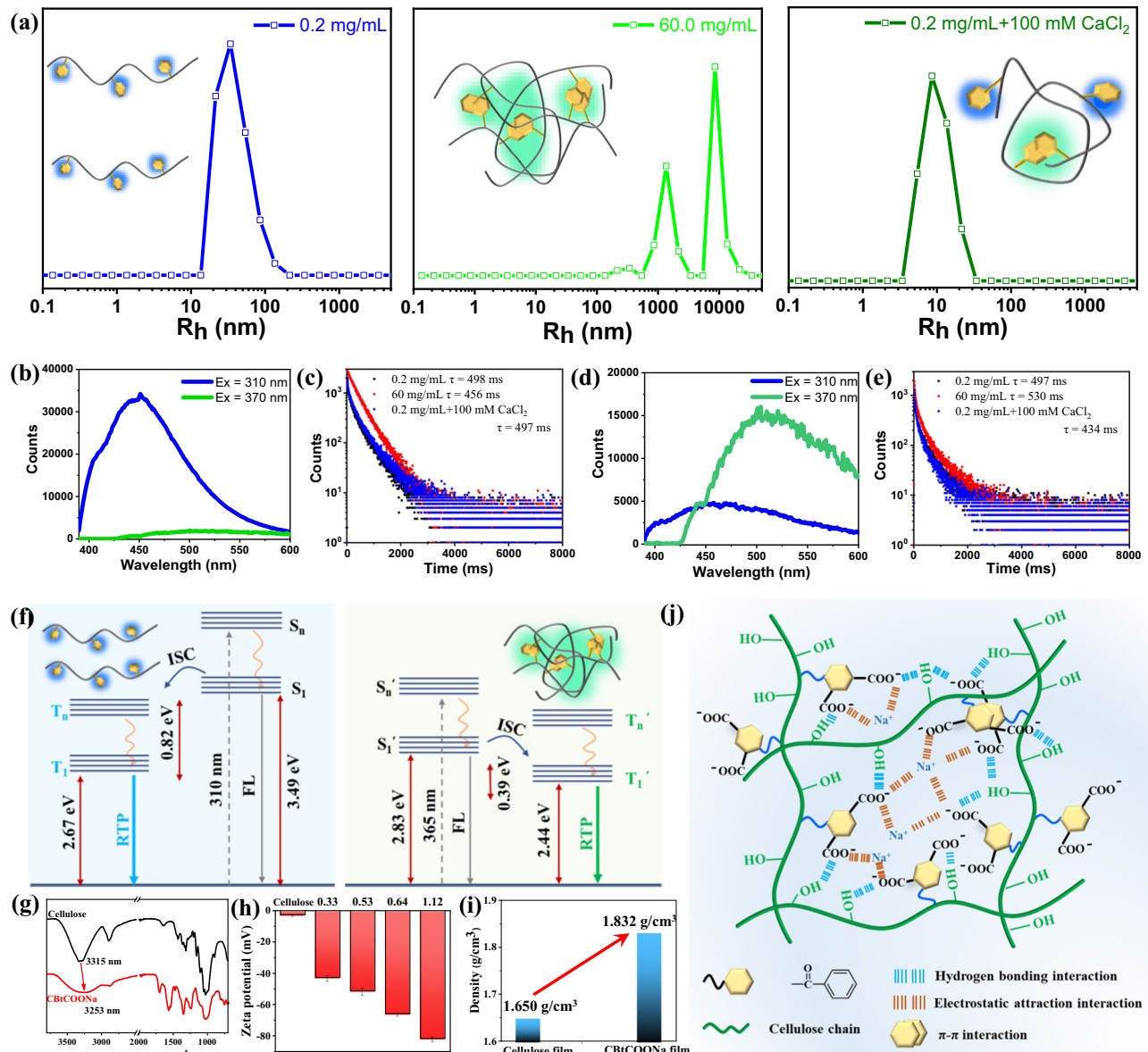

**Fig. 3 | Phosphorescence mechanism of RTP materials. a** DLS of CBtCOONa aqueous solutions with different concentrations; **b** Phosphorescence spectra of 0.2 mg/mL CBtCOONa at 77 K (Ex = 310, 370 nm, delay time = 0.1 ms); **c** Phosphorescence lifetime of 0.2 mg/mL CBtCOONa, 60 mg/mL CBtCOONa, and 0.2 mg/mL CBtCOONa with 100 mM CaCl₂ at 77 K (Ex = 310 nm; detection wavelength = 450 nm); **d** Phosphorescence spectra of 60 mg/mL CBtCOONa at 77 K (Ex = 310, 370 nm, delay time = 0.1 ms); **e** Phosphorescence lifetime of 0.2 mg/mL CBtCOONa, 60 mg/mL CBtCOONa, and 0.2 mg/mL CBtCOONa with 100 mM CaCl₂ at 77 K (Ex = 370 nm; detection wavelength = 500 nm); **f** Energy gap of CBtCOONa (DS = 0.54) with different aggregation states; **g** FTIR spectra of CBtCOONa and cellulose; **h** Zeta potential of 1.0 mg/mL CBtCOONa and cellulose; **i** Density of CBtCOONa and cellophane; **j** Schematic illustration of the phosphorescence mechanism.

originates from molecularly-dispersed BtCOONa, and green phosphorescence originates from aggregated BtCOONa (Fig. 3f).

Additionally, the O–H peak for CBtCOONa shifted to a lower wavenumber compared with the O–H stretching vibration peak for natural cellulose, indicating that the stronger hydrogen bonding interactions were formed in CBtCOONa (Fig. 3g). Zeta potential gradually increased as the DS of CBtCOONa increased from 0.33 to 1.12. Hence, the electrostatic interactions between the chains are became increasingly strong in CBtCOONa (Fig. 3h). The density of CBtCOONa film was 1.832 g·cm⁻³, which was considerably higher than that of the cellulose film (1.650 g·cm⁻³), illustrating that the CBtCOONa chains were tightly packed (Fig. 3i). Overall, the molecularly-dispersed and aggregate states of the BtCOONa group promoted inter-system crossing. The strong hydrogen-bonding and electrostatic attraction

interactions facilitated the tight packing of CBtCOONa chains, which effectively suppressed the non-radiative transition (Fig. 3j). Therefore, various RTP materials with different emission modes were obtained via regulating the aggregation state of CBtCOONa. The molecularly-dispersed CBtCOONa exhibited blue phosphorescence; the aggregated CBtCOONa emitted green phosphorescence; and the CBtCOONa with a coexistence state of single molecular chains and aggregates had color-tunable phosphorescence with a strong excitation-dependence. Such an aggregation-regulated principle provides a facile strategy to prepare the proof-of-concept and on-demand RTP materials by using a single organic component.

In order to prove the universality of the above strategy, three cellulose derivatives have been synthesized and used, including cellulose phthalate sodium (CPhCOONa), carboxymethylcellulose

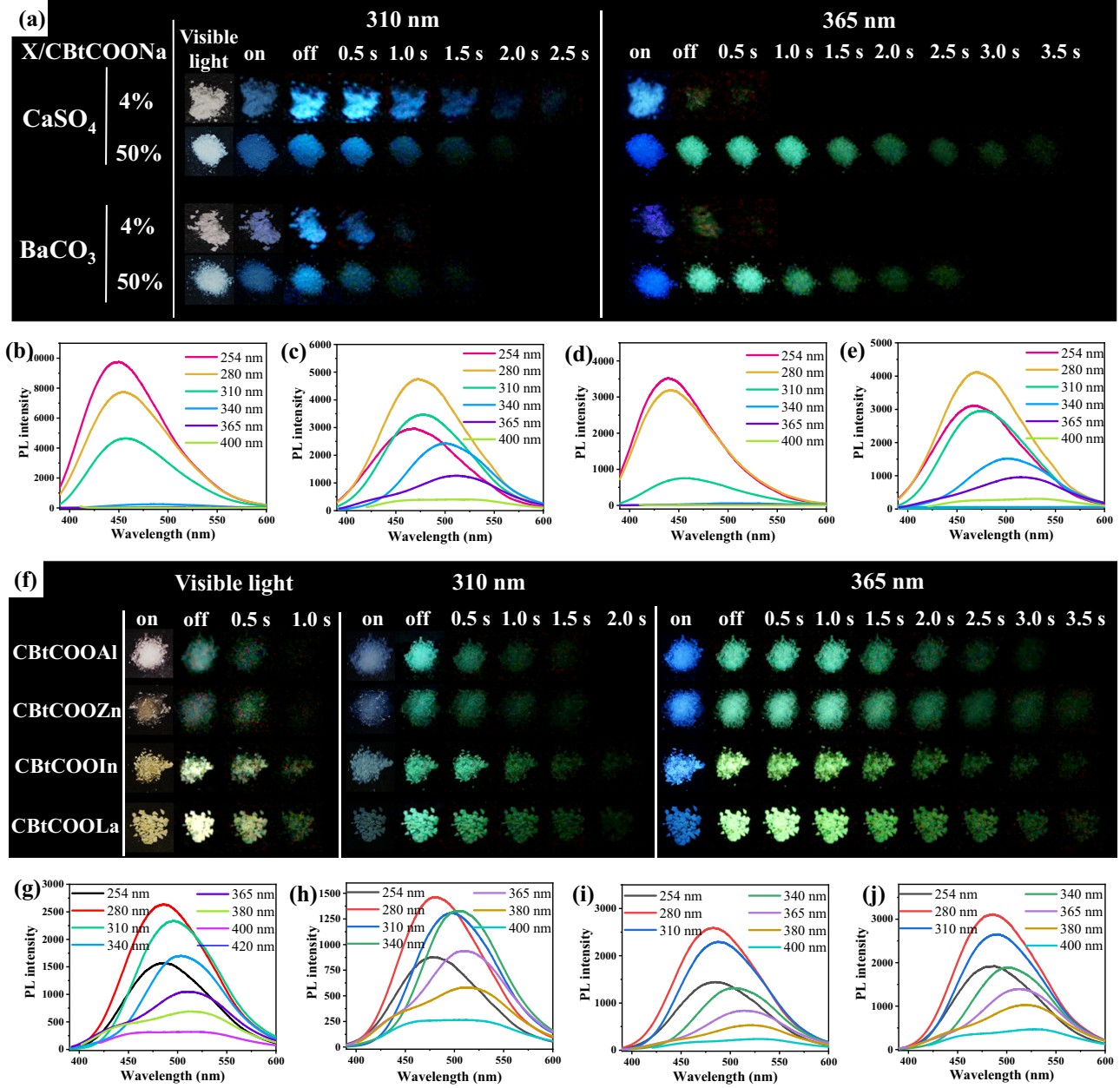

**Fig. 4 | RTP properties of different samples. a** Photographs of CBtCOONa/CaSO$_4$ and CBtCOONa/BaCO$_3$ with different CBtCOONa contents under irradiation with lamps and with the lamps off; **b** Phosphorescent spectra of CBtCOONa/CaSO$_4$ with 4% CBtCOONa; **c** Phosphorescent spectra of CBtCOONa/CaSO$_4$ with 50% CBtCOONa; **d** Phosphorescent spectra of CBtCOONa/BaCO$_3$ with 4% CBtCOONa; **e** Phosphorescent spectra of CBtCOONa/BaCO$_3$ with 50% CBtCOONa; **f** Photographs of CBtCOOAl, CBtCOOZn, CBtCOOIn, and CBtCOOLa under irradiation with lamps and with the lamps off; **g** Phosphorescent spectra of CBtCOOAl; **h** Phosphorescent spectra of CBtCOOZn; **i** Phosphorescent spectra of CBtCOOIn; **j** Phosphorescent spectra of CBtCOOLa.

sodium (CMC) and cellulose 1-cyanomethylimidazolium chloride (Cell-ImCNCl). Via using the aggregation-regulated strategy, three series of ultralong RTP materials with multi-mode emission were fabricated successfully (Figs. S9–S11).

### Adjustment of phosphorescence performance

We also used other inorganic salts, such as CaSO$_4$ and BaCO$_3$, to replace CaCO$_3$. The obtained RTP materials exhibited similar phosphorescence performance. When the CBtCOONa content was 4%, CBtCOONa/CaSO$_4$ and CBtCOONa/BaCO$_3$ emitted blue phosphorescence at 310 nm excitation, and no phosphorescence at 365 nm excitation (Fig. 4a). The phosphorescence emission peak changed slightly as the excitation wavelength increased (Fig. 4b, d). When the CBtCOONa content was increased to 50%, CBtCOONa/CaSO$_4$ and

CBtCOONa/BaCO$_3$ emitted blue phosphorescence at 310 nm excitation, and green phosphorescence at 365 nm excitation (Fig. 4a). The phosphorescence emission was obviously excitation-dependent (Fig. 4c, e).

Notably, CBtCOONa exhibited visible-light excitation phosphorescence (Figs. S12 and S13), which can be attributed to the aggregate state of CBtCOONa. As the number of the aggregates increased, the visible-light excitation phosphorescence became increasingly strong (Fig. S14). We further exchanged the cation to prepare a series of cellulose trimellitates with different metal cations (CBtCOOM). The resultant cellulose trimellitates, i.e., CBtCOOAl, CBtCOOZn, CBtCOOIn, and CBtCOOLa exhibited excellent phosphorescence properties. They emitted blue fluorescence and bright green phosphorescence at 365 nm excitation. The green phosphorescence lasted

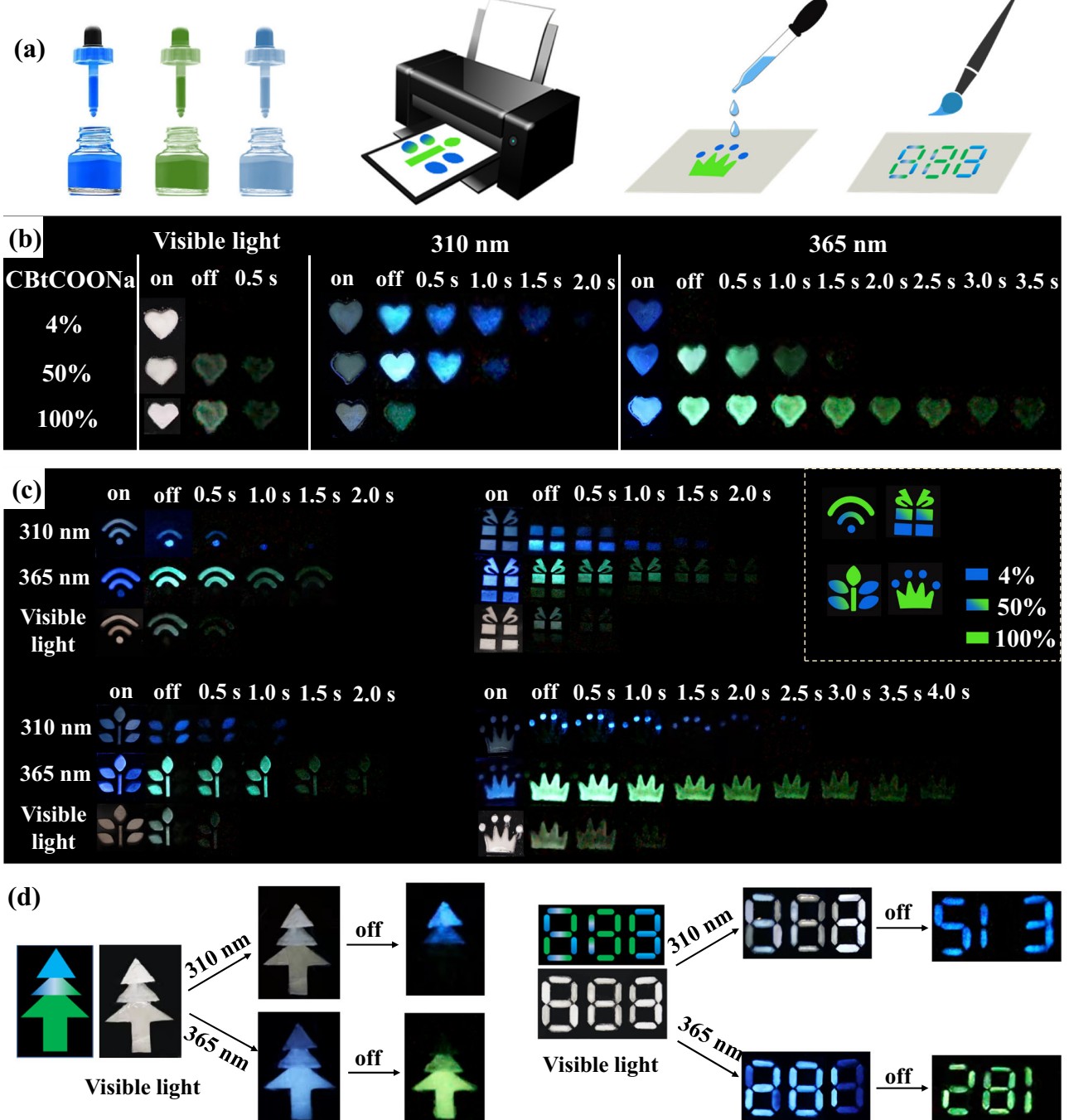

**Fig. 5 | Application of RTP materials in anticounterfeiting and information encryption. a** Applications of phosphorescent inks; **b** Photographs of the phosphorescent patterns; **c** Photographs of the phosphorescent patterns with three inks; **d** Anticounterfeiting pattern and code with three inks.

up to 3.5 s, as observed by the naked eye (Fig. 4f and S15–S17). At visible-light excitation, they emitted phosphorescence. The phosphorescence was light green for CBtCOOAl and CBtCOOZn. The phosphorescence was light yellow for CBtCOOIn and CBtCOOLa (Fig. 4f and S17). The phosphorescence lasted up to 1 s. At 400 nm excitation, CBtCOOAl and CBtCOOZn had broad phosphorescence emission peaks at 510 nm, while the phosphorescence peaks of CBtCOOIn and CBtCOOLa were red-shifted (Fig. 4g–j), because the larger $In^{3+}$ and $La^{3+}$ compared to $Al^{3+}$ and $Zn^{2+}$ formed larger aggregates with $CBtCOO^-$ chains via electrostatic interactions[42–44]. The phosphorescence emission of CBtCOOAl, CBtCOOZn, CBtCOOIn and CBtCOOLa had a certain degree of excitation wavelength dependence

(Fig. 4g–j). Therefore, via changing the cation species, the aggregation state can be adjusted to improve the phosphorescence performance at 365–420 nm excitation.

## Application of RTP materials
Based on the excellent processability and formability of cellulose derivatives (Fig. S18), the above RTP materials can be used as waterborne inks to directly prepare various phosphorescence patterns by inkjet printing and screen printing (Fig. 5). In order to demonstrate this, we selected CBtCOONa/CaCO₃ with 4% CBtCOONa, CBtCOONa, and CBtCOONa/CaCO₃ with 50% CBtCOONa as the blue phosphorescence ink, green phosphorescent ink, and excitation-dependence

phosphorescent ink, respectively (Fig. 5a). When these three inks were used individually, patterns with different phosphorescence colors and lifetimes were observed (Fig. 5b). When all the three inks were used in the same pattern, complex phosphorescent patterns were fabricated (Fig. 5c). At different excitation wavelengths, they would exhibit different phosphorescent pictures. For example, in the tree pattern, only four leaves on two sides of the trunk appear in blue color at 310 nm excitation. At 365 nm excitation, the trunk, top leaf, and two leaves on the left side appear in green color. Obviously, these RTP materials can exhibit a complicated variation using only a simple pattern, making them a fabulous system for sophisticated anticounterfeiting and information encryption. We made the code with the aforementioned three inks (Fig. 5d and S19). The resultant pattern shows a blue colored number 513 at 310 nm excitation, and changes to a green colored number 281 at 365 nm excitation. In addition, CBtCOONa film shows reversible water/heating-responsive RTP effect (Fig. S20), because of the destructive effect of water to the intermolecular interactions[16]. More significantly, the raw materials used to prepare multimode RTP materials are natural materials that are completely biodegradable, nontoxic and low-cost. Thus, these RTP inks do not cause environment pollution even when they are discarded in the environment after use.

## Discussion

A principle, the aggregation-regulated phosphorescence, was demonstrated to develop high-performance RTP materials. Accordingly, we developed eco-friendly ultralong RTP materials with multimode emission, adjustable excitation-dependence and visible-light excitation, via using a single biopolymer component, cellulose trimellitate. Three RTP materials with different emission modes were fabricated: blue, green, and color-tunable RTP materials with a strong excitation-dependence. In addition, the aggregated cellulose trimellitates exhibited appealing visible-light excitation phosphorescence. Notably, the raw materials used to prepare the multimode RTP materials were natural and entirely environmentally-friendly. Therefore, the resultant multi-mode RTP materials were used as advanced anticounterfeiting inks to successfully prepare various sustainable complex phosphorescence patterns and information codes successfully, indicating a huge potential in anti-counterfeiting, information encryption, intelligent labels, etc. This work provides a facile and general strategy to prepare proof-of-concept RTP materials and deepens the understanding of the luminescence mechanism and color regulation of polymer materials.

## Methods

### Synthesis of cellulose trimellitate (CBtCOOH)

Four grams (24.69 mmol) of cellulose was completely dissolved in 76 g of the ionic liquid 1-allyl-3-methylimidazolium chloride (AmimCl) at 80 °C. Then, 4-dimethylaminopyridine (DMAP) (300 mg, 2.45 mmol) and trimellitic anhydride (9.48–14.22 g, 49.38–74.07 mmol) were added into the cellulose/AmimCl solution at 80 °C for 12 h. Thereafter, ethanol was added into the reaction system to remove unreacted trimellitic anhydride. The reaction solution was precipitated in ethanol (400 mL) with 1 mL of concentrated hydrochloric acid. The precipitate was filtered and washed thrice with an ethanol solution and dried under vacuum for 24 h. The DS of obtained CBtCOOH was calculated from $^1$H-NMR spectrum. $^1$H-NMR (400 MHz, DMSO-$d_6$): δ 7.50–8.50 (m, 3H), 2.70–5.50 (m, 6H); IR (Nujol): 3400 cm$^{-1}$ (O-H), 1703 cm$^{-1}$ (C = O), 1612 cm$^{-1}$ (C = C), 1494 cm$^{-1}$ (C = C), 754 cm$^{-1}$ (=C−H).

### Synthesis of sodium cellulose trimellitate (CBtCOONa)

Two grams (DS = 0.54, 7.35 mmol) of CBtCOOH was dissolved in 50 mL of double-distilled water. Then, NaHCO$_3$ (1.26 g, 15.00 mmol) was added to the solution. After stirring for 12 h, the solution was centrifuged. The supernatants were transferred to a dialysis bag. Ultrapure water was used as the dialysate. The final product was obtained via freeze-drying.

### Synthesis of RTP materials with different CBtCOONa contents

CBtCOONa (DS = 0.54) aqueous solution with different concentrations (0.2, 2.0, 20.0, 60.0 and 80.0 mg/mL), Na$_2$CO$_3$ (1.0 mmol/mL), CaCl$_2$ (1.0 mmol/mL), Na$_2$SO$_4$ (1.0 mmol/mL) and BaCl$_2$ (1.0 mmol/mL) were prepared. Considering CBtCOONa/CaCO$_3$ as an example.

### CBtCOONa/CaCO$_3$

Four sample vials were prepared. CBtCOONa aqueous solution (2.0 mL, 0.2–80.0 mg/mL) was added into each vial. Then, the Na$_2$CO$_3$ (0.5–1.6 mL, 1.0 mmol/mL) was added into each vial dropwise. After stirring for 10 min, the CaCl$_2$ (0.5–1.6 mL, 1.0 mmol/mL) was added into each vial dropwise. After a strong stirring for 30 min, the mixed suspended solution was transferred to a dialysis bag. Ultrapure water was used as the dialysate. After a dialysis for 12 h, the solution in the dialysis bag was dried at 80 °C with stirring. The final products were dried under vacuum at 60 °C for 24 h. The contents of cellulose derivatives were calculated according to the mass of dissolved CBtCOONa and CaCO$_3$ produced from Na$_2$CO$_3$ and CaCl$_2$.

### Synthesis of CBtCOOM

Four sample vials were prepared. Then, 0.31 g of CBtCOOH (DS = 0.54, 1.00 mmol) and 10 mL of ultrapure water were put into each vial. AlCl$_3$ (67 mg, 0.50 mmol), ZnCl$_2$ (109 mg, 0.80 mmol), InCl$_3$ (111 mg, 0.50 mmol) and LaCl$_3$ (123 mg, 0.50 mmol) were added into each vial, respectively. After stirring for 12 h, the white precipitates were centrifuged and washed with water thrice. The final products were dried under vacuum at 60 °C for 24 h to obtain CBtCOOM.

### Synthesis of BtCOONa

Trimellitic anhydride (1.92 g, 10 mmol) and NaOH (1.20 g, 30 mmol) were added into 20 mL distilled water. After stirring for 30 min, the transparent solution was dried under vacuum at 60 °C for 24 h to obtain BtCOONa.

### Preparation of RTP patterns

The CBtCOONa/CaCO$_3$ with 4% CBtCOONa aqueous suspension was used as the blue phosphorescence ink. The CBtCOONa aqueous solution with the CBtCOONa concentration of 3 wt% was used as the green phosphorescent ink. The CBtCOONa/CaCO$_3$ with 50% CBtCOONa aqueous suspension was used as the excitation-dependence phosphorescent ink. Those inks were directly used to prepare patterns by inkjet printing and screen printing. For the complex phosphorescence patterns, different areas were printed with different kinds of inks (Fig. 5 and S19). Blue area represents CBtCOONa/CaCO$_3$ with 4% CBtCOONa ink, green area represents CBtCOONa ink, blue and green area represents CBtCOONa/CaCO$_3$ with 50% CBtCOONa ink.

## Data availability

All relevant data are included in this Article and its Supplementary Information files.

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

## Acknowledgements

This work was supported by the National Natural Science Foundation of China (No. 52173292 and U2004211) (J.M.Z. and J.Z.) and Youth Innovation Promotion Association CAS (No. 2018040) (J.M.Z.).

## Author contributions

J.X.Y. and J.M.Z. conceived the idea. J.X.Y. performed the experiments. X.Z., Q.Y.N., Y.R.W., K.F.J., and C.C.Y. offered help to J.X.Y. for the experiments. J.X.Y., J.M.Z., Z.Y.Y., and J.Z. discussed the results and wrote the manuscript.

## Competing interests

The authors declare no competing interests.
