## [Peer Review File · Nature Communications]

Aggregation-Regulated Room-Temperature Phosphorescence Materials with Multi-Mode Emission, Adjustable Excitation-Dependence and Visible-Light ExcitationREVIEWER COMMENTS

Reviewer #1 (Remarks to the Author):

Focusing on room-temperature phosphorescence (RTP), Zhang et al. reported a simple method to construct eco-friendly RTP materials based on single-component cellulose trimellitate. Different from previous works by precise chemical synthesis and structural modifications of phosphors, they discovered that the RTP properties of these cellulose trimellitates could be manipulated by regulating the aggregate state. Through CaCO₃ precipitation and solvent evaporation methods, they prepared three kinds of samples with different states. The molecularly separated CBtCOONa exhibited blue phosphorescence, while the aggregated CBtCOONa displayed green phosphorescence. And the CBtCOONa with two states exhibited two RTP colors with the excitation-dependent property. The origin of RTP emission was also confirmed through the model compound of sodium trimellitate (BtCOONa). In addition, by changing the inorganic salts, the emission color could be slightly adjusted. The work seems to be carefully carried out, and the results are interesting for publication. However, the breakthrough is moderate, and several comments need to be clarified and considered. Hence, a major revision is suggested before considering it for publication.

1)The authors mentioned that the degree of substitution (DS) of CBtCOOH was 0.54. However, the synthetic method indicated that they used different molars of trimellitic anhydride (9.48-14.22 g, 49.38-74.07 mmol). Hence, did the author optimize the reaction to obtain the DS? Whether the products with different molar ratios of trimellitic anhydride could show different RTP performance?

2)The molecularly separated CBtCOONa and the coexistent state of CBtCOONa were fabricated by immobilizing in CaCO₃. And the authors also mentioned that the RTP origin is the sodium trimellitate. So, I am curious about the role of cellulose in this system. Is it possible to obtain similar RTP materials using the same method in sodium trimellitate solutions instead of cellulose trimellitate?

3)In the abstract, the authors claimed the method proposed in this work is a general strategy to obtain eco-friendly ultralong RTP materials. However, its universality has not been demonstrated. To check the universality, the author may consider using other RTP units to replace trimellitic anhydride.

4)For Fig. 2, the QYs for all these samples are smaller than 4%, which is a much lower value than other reported RTP systems. The authors may add some comments to it.

5)In a highly dilute solution (0.2 mg/mL), CBtCOONa should exhibit a molecularly dispersed state without aggregation, as shown in Fig. 3a. Usually, no DLS signal can be detected for the dispersed solution. Why are there 2 nm aggregates in this dilute solution?

6)On page 7, the description that "dilute CBtCOONa solutions emitted blue phosphorescence without excitation dependence at 254-310 nm excitation" is misleading. From Fig. S4 and 1b, actually, dilute CBtCOONa solution showed different emission maximum and even green RTP under excitation 365 nm, indicating its excitation-dependent property.

7)For dilute solution and concentrated solution, and metal participated solution of CBtCOONa, the aggregate state and inter/intrachain interactions are totally different. Why did they show almost the same lifetimes of RTP, around 450-498 ns?

8)Calcium carbonate is insoluble in water. How did the author prepare waterborne inks based on

CBtCOONa/CaCO₃ with different ratios of CBtCOONa?

9) For Fig. 5a, this diagram is oversimplified and doesn't show how to fabricate patterns with those inks. Besides, the detailed methods to fabricate these patterns should be described in the Method.

10) For Fig. 4 and 5, What kind of lamp is used as the visible-light source? What is the wavelength?

11) The authors should polish the manuscript carefully, as several grammatical and spelling mistakes are observed. The logic of the manuscript needs to be improved.

Reviewer #2 (Remarks to the Author):

Organic materials with bright and tunable room temperature phosphorescence (RTP) are important for optoelectronic and biomedical applications. Particularly, those from natural sources are even promising considering its biocompatibility, environment-friendliness, and rich sources. In this contribution, by conjugating anionic phenylcarboxylate substituents onto the cellulose chain and further neutralization, the authors constructed the sodium cellulose trimellitate (CBtCOONa) as a kind of RTP material, which demonstrates blue, green, and color-tunable RTP in molecularly-dispersed state, aggregate state, and the coexistence state of the above two, respectively. Noticeably, visible-light excitation was observed from such RTP material as well as other cellulose trimellitates with different metal cations (CBtCOOM). Corresponding characterizations also show the potential of CBtCOONa in advanced anti-counterfeiting inks. Overall, the experimental design and application demo are commendable, and the findings are interesting and important for further exploration of novel RTP materials. Therefore, this reviewer recommends it for publication in Nat. Commun. with some minor revisions.

1. The synthesis of CBtCOOH is proved by ¹H NMR and FTIR in the manuscript. However, these characterizations can only verify the coexistence of the cellulose chain and the small molecule in the system, while whether these two parts are conjugated is not sufficiently proved.
2. Line 163-164, it is claimed that there is no excitation dependence at 254-310 nm excitation when CBtCOONa content is 4%-20%, which seems not accurate. The main peak is slightly red-shifted as the excitation wavelength changes from 254 to 310 nm in Figure S4(a).
3. Line 222, "phosphor" might be "phosphorescence".
4. Line 255, it is claimed that inter-system crossing is promoted. Are there any further evidences for it?
5. Line 303-304, the red-shifted phosphorescence of CBtCOOIn and CBtCOOLa is attributed to larger radii of In³⁺ and La³⁺ and further formed larger aggregates with CBtCOO⁻ chains. Further explanation on this deduction is encouraged.
6. The author discussed multi-color and stimuli-responsive RTP materials in the Introduction part. Some highly related work can be included: DOI: 10.1039/d0cs01087a.

Reviewer #3 (Remarks to the Author):

In this manuscript, the authors reported a new kind of RTP materials with multi-mode emission, adjustable excitation-dependence and visible-light excitation based on cellulose derivatives.

Experimental results showed that the aggregation regulation of the phosphor groups in cellulose should be mainly responsible for these unique characteristics. The results are interesting and can be published after revisions.

1. In this work, the covalent linkage between anionic phenylcarboxylate and cellulose should be important for the efficient RTP effect. To further certify it, the RTP properties of anionic phenylcarboxylate derivatives and cellulose mixtures through physical doping should be measured to make a careful comparison.
2. How about the molecular weights of cellulose and cellulose trimellitate? The measurements of them would be much helpful to calculate the degree of substitution (DS) of CBtCOOH.
3. The poor solubility of cellulose in common solvents has largely limited its practical applications. In this work, the combination of anionic phenylcarboxylate and cellulose led to the relatively good solubility in aqueous solution, which could largely promote the processibility. Thus, the authors are suggested to add some discussions about it.
4. In Figure 1 and 4, the afterglow times of aggregates are always longer than those of single molecular chains. This is very interesting and the authors should make more discussions about it.
5. The obtained cellulose trimellitate showed relatively good solubility in aqueous solution (Figure 3). Then, if its solids show water/heating-responsive RTP effect for the destructive effect of water to the intermolecular interactions?
6. Some important references related to this work should be added, such as *Sci. Adv.* 2022, 8, eabl8392; *Chem. Soc. Rev.*, 2021, 50, 12616; *Chinese J. Chem.*, 2022, 40, 2359; *Acc. Chem. Res.*, 2020, 53, 962 and so on.

Reviewer #4 (Remarks to the Author):

In this manuscript entitled "Aggregation-Regulated Room-Temperature Phosphorescence Materials with Multi-Mode Emission, Adjustable Excitation-Dependence and Visible-Light Excitation", You et al. presented RTP materials with multiple emission and special excitation modes by controlling the aggregation state of CBtCOONa. The strategy is simple, novel, and interesting. It provided a new strategy to fabricate RTP materials. The manuscript is well written and describes very thoroughly the synthesis processes and the applications of the RTP materials. It has a wide readership. It is suitable to the scope of NATURE COMMUNICATIONS. I recommend publishing this paper after minor revisions.

1. Are the RTP materials crystal or amorphous? XRD spectra of the RTP materials should be shown.
2. Except the reported metal ions, what is the effect of other common metal ions, such as Fe²⁺, Fe³⁺, Cu²⁺, Co²⁺ and so on?
3. Cellulose could demonstrate as a chiral polymer. Is the RTP chiral?

Answers to Comments by Reviewers

Answers to Comments by Reviewer # 1

- (1) **Reviewer #1 wrote:** *Focusing on room-temperature phosphorescence (RTP), Zhang et al. reported a simple method to construct eco-friendly RTP materials based on single-component cellulose trimellitate. Different from previous works by precise chemical synthesis and structural modifications of phosphors, they discovered that the RTP properties of these cellulose trimellitates could be manipulated by regulating the aggregate state. Through CaCO₃ precipitation and solvent evaporation methods, they prepared three kinds of samples with different states. The molecularly separated CBtCOONa exhibited blue phosphorescence, while the aggregated CBtCOONa displayed green phosphorescence. And the CBtCOONa with two states exhibited two RTP colors with the excitation-dependent property. The origin of RTP emission was also confirmed through the model compound of sodium trimellitate (BtCOONa). In addition, by changing the inorganic salts, the emission color could be slightly adjusted. **The work seems to be carefully carried out, and the results are interesting for publication.** However, the breakthrough is moderate, and several comments need to be clarified and considered. Hence, a major revision is suggested before considering it for publication.*

Answer: Thanks for your kind comments. We have revised the manuscript as suggestion.

- (2) **Reviewer #1 wrote:** *The authors mentioned that the degree of substitution (DS) of CBtCOOH was 0.54. However, the synthetic method indicated that they used different molars of trimellitic anhydride (9.48-14.22 g, 49.38-74.07 mmol). Hence, did the author optimize the reaction to obtain the DS? Whether the products with different molar ratios of trimellitic anhydride could show different RTP performance?*

Answer: Thanks for your kind suggestion. We have synthesized a series of CBtCOOH with different DS values from 0.33 to 1.12 by controlling the reaction

time and the molar ratio of trimellitic anhydride (BtCOOH) and AGU (Table R1). The quantum yield of CBtCOONa (DS = 0.54) is 3.14% and the RTP lifetime is 240 ms, which is higher than those of other samples (Figure R1). Therefore, we used CBtCOONa with DS = 0.54 as the raw materials. The synthesis results have added in the revised Supplementary Information.

Table R1 Synthesis of CBtCOOH under different reaction conditions.

Samples	Molar ratio of BtCOOH and AGU	Molar ratio of DMAP and AGU	Reaction time/h	Temperature/°C	DS
1	2:1	0.1:1	12	80	0.33
2	2.5:1	0.1:1	12	80	0.54
3	3:1	0.1:1	9	80	0.63
4	3:1	0.1:1	12	80	1.12

Figure R1 RTP performance of CBtCOONa with different DS values. (a) RTP lifetime spectra of CBtCOONa (Ex = 370). (b) Photoluminescence quantum yield of CBtCOONa (Ex = 370 nm; detection wavelength = 510 nm).

(3) **Reviewer #1 wrote:** *The molecularly separated CBtCOONa and the coexistent state of CBtCOONa were fabricated by immobilizing in CaCO₃. And the authors also mentioned that the RTP origin is the sodium trimellitate. So, I am curious about the role of cellulose in this system. Is it possible to obtain similar RTP*

materials using the same method in sodium trimellitate solutions instead of cellulose trimellitate?

Answer: When the sodium trimellitate (BtCOONa) was chose as the raw materials and the same preparation method was used, we obtained only two types of RTP materials, which are green RTP material with a weak excitation-dependence and green/blue RTP material with a strong excitation-dependence (Figure R2a). The RTP material with only blue phosphorescence was unable to be prepared. Because the molecularly dispersed state of trimellitate group is difficultly obtained, if there is not a polymer chain to immobilize the trimellitate group. In addition, the RTP lifetime and quantum yield of CBtCOONa were higher than those of BtCOONa (Figure R2b and Figure R3a-c). Furthermore, CBtCOONa could be excited with visible light, while the BtCOONa exhibited no phosphorescence (Figure R2b).

Therefore, cellulose exhibits three obvious advantages. Firstly, cellulose chain immobilizes the trimellitate group to suppress the phosphor motion and separate the phosphor. The molecularly-dispersed state of the trimellitate group can be obtained based on the synergistic effect of the anchoring and diluting effect of the cellulose backbone. So, the RTP material with only blue phosphorescence can be obtained. Secondly, along the cellulose chains, there are numerous hydroxyl groups, which provide strong hydrogen-bonding interactions. So, the molecular motion has been inhibited effectively. The obtained RTP materials have better phosphorescence performance than the BtCOONa. Finally, cellulose derivatives have excellent processability and formability. CBtCOONa can be directly processed into flexible film, coating, and patterns (Figure R3d).

Figure R2 (a) Photographs of RTP materials with different BtCOONa contents under irradiation with 310 and 365 nm lamps and with the lamps off. (b) Contrast of pure CBtCOONa and pure BtCOONa with different lamps and with lamps off.

Figure R3 (a) Quantum yield of BtCOONa and CBtCOONa; (b) Phosphorescence intensity of BtCOONa and CBtCOONa at 340 nm excitation; (c) RTP lifetime of BtCOONa and CBtCOONa; (d) Photographs of CBtCOONa film.

(4) **Reviewer #1 wrote:** *In the abstract, the authors claimed the method proposed in this work is a general strategy to obtain eco-friendly ultralong RTP materials. However, its universality has not been demonstrated. To check the universality, the author may consider using other RTP units to replace trimellitic anhydride.*

Answer: Thanks for your kind comments. In order to prove the universality, three new cellulose derivatives have been synthesized and used, including cellulose phthalate sodium (CPhCOONa), carboxymethylcellulose sodium (CMC) and cellulose 1-cyanomethylimidazolium chloride (Cell-ImCNCI). Via using our strategy, three new series of ultralong RTP materials with multi-mode emission were fabricated, as shown in Figures R4-R6. Thus, the strategy provided in this manuscript is universal to prepare the novel RTP materials.

Figure R4 (a) Photographs of 50 mg/mL CPhCOONa aqueous solution at 77 K taken under 310 nm and 365 nm lamps and with the lamps off, respectively. (b) Photographs of RTP materials with 50% and 100% CPhCOONa contents under irradiation with 310 and 365 nm lamps and with the lamps off.

Figure R5 (a) Photographs of 30 mg/mL CMC aqueous solution at 77 K taken under 310 nm and 365 nm lamps and with the lamps off, respectively. (b) Photographs of RTP materials with 50% and 100% CMC contents under irradiation with 310 and 365 nm lamps and with the lamps off.

Figure R6 (a) Photographs of 50 mg/mL Cell-ImCNCI aqueous solution at 77 K taken under 310 nm and 365 nm lamps and with the lamps off, respectively. (b) Photographs of RTP materials with 50% and 100% Cell-ImCNCI contents under irradiation with 310 and 365 nm lamps and with the lamps off.

- (5) **Reviewer #1 wrote:** *For Fig. 2, the QYs for all these samples are smaller than 4%, which is a much lower value than other reported RTP systems. The authors may add some comments to it.*

Answer: Thanks for your kind suggestion. The quantum yield of BtCOONa is only 2.91%. After being immobilized on cellulose chains, the quantum yield of obtained CBtCOONa (DS = 0.54) increases to 3.41% (Figure R3a). The ISC effect can be promoted via the heavy-atom effect, molecular aggregation, lone-pair electron incorporation, energy-gap narrowing, and so on (*Nat. Chem.* 2011, 3, 205-210; *Chem. Sci.* 2017, 8, 590-599; *Adv. Opt. Mater.* 2019, 7, 1800820). We will introduce heavy-atom to promote the quantum yield in our further work.

- (6) **Reviewer #1 wrote:** *In a highly dilute solution (0.2 mg/mL), CBtCOONa should exhibit a molecularly dispersed state without aggregation, as shown in Fig. 3a. Usually, no DLS signal can be detected for the dispersed solution. Why are there 2 nm aggregates in this dilute solution?*

Answer: Thanks for your kind suggestion. The hydrodynamic radius (R_h) of single polymer chain is in a range of 10-50 nm, which depends on the molecular weight of polymer. As the concentration increases, polymer chains form the aggregate, the

R_h of which is larger than 50 nm. The similar phenomena have been reported. For example, the R_h of cellulose in dilute solution is around 20 nm (Figure R7) (*Green Chem.*, 2022, 24, 3824-3833; *Polymer*, 2001, 42, 6765-6773; *Green Chem.*, 2022, 24, 3824-3833; *Carbohydr. Polym.* 2014, 112, 125-133; *Phys. Chem. Chem. Phys.*, 2022, 00, 1-3). As the concentration increases, the cellulose chains form the aggregate, the R_h of which is near or greater than 100 nm.

Figure R7 Hydrodynamic radius (R_h) distribution of cellulose in ionic liquids at 25 °C
(*Green Chem.*, 2022, 24, 3824-3833).

- (7) **Reviewer #1 wrote:** *On page 7, the description that “dilute CBtCOONa solutions emitted blue phosphorescence without excitation dependence at 254-310 nm excitation” is misleading. From Fig. S4 and 1b, actually, dilute CBtCOONa solution showed different emission maximum and even green RTP under excitation 365 nm, indicating its excitation-dependent property.*

Answer: Thanks for your kind comments. We have corrected the description in the manuscript as suggestion.

- (8) **Reviewer #1 wrote:** *For dilute solution and concentrated solution, and metal participated solution of CBtCOONa, the aggregate state and inter/intrachain interactions are totally different. Why did they show almost the same lifetimes of RTP, around 450-498 ms?*

Answer: The lifetime of dilute solution, concentrated solution and metal

participated solution of CBtCOONa was measured at 77 K, which means that the hydrogen bonding interactions between H₂O and CBtCOONa chains predominated the micro-environments. The hydrogen bonding interactions are similar in the dilute solution, concentrated solution and metal participated solution of CBtCOONa. Therefore, the similar RTP lifetimes were obtained.

(9) **Reviewer #1 wrote:** *Calcium carbonate is insoluble in water. How did the author prepare waterborne inks based on CBtCOONa/CaCO₃ with different ratios of CBtCOONa?*

Answer: Thanks for your kind comments. The CBtCOONa/CaCO₃ aqueous suspension was prepared and used as the inks. In the CBtCOONa aqueous solution, the Na₂CO₃ aqueous solution and CaCl₂ aqueous solution were added sequentially. Under a strong stirring, the CBtCOONa/CaCO₃ aqueous suspension was obtained. The detailed method was shown in the manuscript.

(10) **Reviewer #1 wrote:** *For Fig. 5a, this diagram is oversimplified and doesn't show how to fabricate patterns with those inks. Besides, the detailed methods to fabricate these patterns should be described in the Method.*

Answer: Thanks for your kind suggestion. We have revised the diagram about how to fabricate patterns in our manuscript. Moreover, the detailed methods are added in the Method. In addition, the design for the complex RTP pattern was shown in Figure R8, and was added in the Supplementary Information.

Figure R8 Design schematic for a complex RTP pattern.

(11) **Reviewer #1 wrote:** *For Fig. 4 and 5, What kind of lamp is used as the visible-*

light source? What is the wavelength?

Answer: Thanks for your kind suggestion. We used 8 W flashlight and the wavelength is 390-780 nm.

(12) Reviewer #1 wrote: *The authors should polish the manuscript carefully, as several grammatical and spelling mistakes are observed. The logic of the manuscript needs to be improved.*

Answer: Thanks for your kind suggestion. The whole manuscript has been revised by a native English professor.

Answers to Comments by Reviewer # 2

(1) Reviewer #2 wrote: *Organic materials with bright and tunable room temperature phosphorescence (RTP) are important for optoelectronic and biomedical applications. Particularly, those from natural sources are even promising considering its biocompatibility, environment-friendliness, and rich sources. In this contribution, by conjugating anionic phenylcarboxylate substituents onto the cellulose chain and further neutralization, the authors constructed the sodium cellulose trimellitate (CBtCOONa) as a kind of RTP material, which demonstrates blue, green, and color-tunable RTP in molecularly-dispersed state, aggregate state, and the coexistence state of the above two, respectively. Noticeably, visible-light excitation was observed from such RTP material as well as other cellulose trimellitates with different metal cations (CBtCOOM). Corresponding characterizations also show the potential of CBtCOONa in advanced anti-counterfeiting inks. **Overall, the experimental design and application demo are commendable, and the findings are interesting and important for further exploration of novel RTP materials.** Therefore, this reviewer recommends it for publication in *Nat. Commun.* with some minor revisions.*

Answer: Thanks for your nice comments and recommendation.

(2) Reviewer #2 wrote: *The synthesis of CBtCOOH is proved by ¹H NMR and FTIR in the manuscript. However, these characterizations can only verify the*

coexistence of the cellulose chain and the small molecule in the system, while whether these two parts are conjugated is not sufficiently proved.

Answer: Thanks for your kind suggestion. We mixed the cellulose and sodium trimellitate (BtCOONa) to detect the FTIR. In the mixture of cellulose and BtCOONa, there is a peak at 1590 cm^{-1} (Figure R9), which belongs to the C=O peak of COONa. In the FTIR spectrum of CBtCOONa, there is a new peak 1712 cm^{-1} which is the C=O peak of the ester group. Thus, the BtCOONa was immobilized on cellulose chain by the ester bond.

In addition, the cellulose/BtCOONa mixture exhibited weaker phosphorescence performance than CBtCOONa, as shown in Figure R10.

Figure R9 FTIR spectra of MCC, BtCOONa, MCC/BtCOONa mixture and CBtCOONa.

Figure R10 Photographs of CBtCOONa and cellulose/BtCOONa mixture under irradiation with lamps and with the lamps off.

- (3) **Reviewer #2 wrote:** *Line163-164, it is claimed that there is no excitation dependence at 254-310 nm excitation when CBtCOONa content is 4%-20%, which seems not accurate. The main peak is slightly red-shifted as the excitation wavelength changes from 254 to 310 nm in Figure S4(a).*

Answer: Thanks for your kind suggestion. We have corrected the description in the manuscript.

- (4) **Reviewer #2 wrote:** *Line 222, “phosphor” might be “phosphorescence”.*

Answer: Thanks for your kind suggestion. We have revised in the manuscript.

- (5) **Reviewer #2 wrote:** *Line 255, it is claimed that inter-system crossing is promoted. Are there any further evidences for it?*

Answer: Thanks for your kind suggestion. The RTP lifetime and quantum yield of CBtCOONa are better than those of BtCOONa (Figure R11 and R12), indicating that the ISC process has been promoted after bonding BtCOONa onto cellulose chain.

Figure R11 Contrast of pure CBtCOONa and pure BtCOONa with different lamps and with lamps off.

Figure R12 (a) Quantum yield of BtCOONa and CBtCOONa; (b) Phosphorescence intensity of BtCOONa and CBtCOONa at 340 nm excitation; (c) RTP lifetime of BtCOONa and CBtCOONa; (d) Photographs of CBtCOONa film.

(6) **Reviewer #2 wrote:** Line 303-304, the red-shifted phosphorescence of CBtCOONa and CBtCOOLa is attributed to larger radii of In^{3+} and La^{3+} and further formed larger aggregates with CBtCOO^- chains. Further explanation on this deduction is encouraged.

Answer: Thanks for your nice suggestion. We have cited some relative articles to explain this deduction in the manuscript (*Angew. Chem. Int. Ed.* 2018, 57, 678-682; *Nat. Commun.* 2019, 10, 4247; *Sci. China. Chem.* 2022, 66, 367-387).

(7) **Reviewer #2 wrote:** The author discussed multi-color and stimuli-responsive RTP materials in the Introduction part. Some highly related work can be included: DOI: 10.1039/d0cs01087a.

Answer: Thanks for your kind suggestion. We have cited the highly-related work in the revised manuscript.

Answers to Comments by Reviewer # 3

- (1) **Reviewer #3 wrote:** *In this manuscript, the authors reported a new kind of RTP materials with multi-mode emission, adjustable excitation-dependence and visible-light excitation based on cellulose derivatives. Experimental results showed that the aggregation regulation of the phosphor groups in cellulose should be mainly responsible for these unique characteristics. **The results are interesting and can be published after revisions.***

Answer: Thanks for your nice comments and recommendation.

- (2) **Reviewer #3 wrote:** *In this work, the covalent linkage between anionic phenylcarboxylate and cellulose should be important for the efficient RTP effect. To further certify it, the RTP properties of anionic phenylcarboxylate derivatives and cellulose mixtures through physical doping should be measured to make a careful comparison.*

Answer: Thanks for your kind suggestion. In order to obtain the homogenous cellulose/BtCOONa mixture, the cellulose hydrogel prepared from ionic liquid was soaked in BtCOONa aqueous solution for 5 h. Then, after drying under vacuum at 60 °C for 24 h, the cellulose/BtCOONa mixture was obtained. It exhibited weaker phosphorescence performance than CBtCOONa, as shown in Figure R13. The RTP intensity of the cellulose/BtCOONa mixture is much weaker than that of CBtCOONa with 365 nm lamp off. Moreover, the cellulose/BtCOONa mixture does not exhibit RTP property after the visible light is turned off. These above phenomena illustrated the importance of the covalent linkage between anionic phenylcarboxylate and cellulose.

Figure R13 Photographs of CBtCOONa and cellulose/BtCOONa mixture under irradiation

with lamps and with the lamps off.

- (3) **Reviewer #3 wrote:** *How about the molecular weights of cellulose and cellulose trimellitate? The measurements of them would be much helpful to calculate the degree of substitution (DS) of CBtCOOH.*

Answer: Thanks for your kind suggestion. The weight-average molecular weight (M_w) of MCC is 45360, and the degree of polymerization (DP_w) of MCC is 280 measured by LS detection (*Anal. Chem.* **2022**, *94*, 5432-5440). The M_w of MCC is 72090, and the DP_w of MCC is 445 measured by RI detection. The M_w of CBtCOONa is 86403, and the DP of CBtCOONa is 321 measured by RI detection. However, the mobile phases of MCC and CBtCOONa are ionic liquid and H_2O , respectively, so the results are incomparable. It is worth noting that the DS of CBtCOOH was calculated by 1H -NMR, which is an accepted method.

- (4) **Reviewer #3 wrote:** *The poor solubility of cellulose in common solvents has largely limited its practical applications. In this work, the combination of anionic phenylcarboxylate and cellulose led to the relatively good solubility in aqueous solution, which could largely promote the processibility. Thus, the authors are suggested to add some discussions about it.*

Answer: Thanks for your kind suggestion. Indeed, CBtCOONa has an excellent solubility in water. The CBtCOONa aqueous solution with 4 wt% concentration is homogenous and transparent (Figure R14a), indicating the good solubility of CBtCOONa. Subsequently, CBtCOONa can be processed into films, coatings and patterns by conventional methods. For example, the flexible CBtCOONa film can be easily obtained by drying the CBtCOONa aqueous solution (Figure R14b).

Figure R14 (a) Photograph of CBtCOONa aqueous solution with 4 wt% concentration; (b) Photographs of CBtCOONa film.

(5) **Reviewer #3 wrote:** *In Figure 1 and 4, the afterglow times of aggregates are always longer than those of single molecular chains. This is very interesting and the authors should make more discussions about it.*

Answer: Thanks for your kind suggestion. The more efficient ISC observed in aggregates is ascribed to possible ISC channels formed thanks to energy-level splitting. When moving from monomer to aggregate, substantial electronic interactions among chromophores cause overlap between the excitonic orbitals and, thus, give rise to energy splitting (Figure R15). The splitting can generate more ISC channels and potentially boost ISC and RTP efficiency in aggregates (*Nat. Rev. Mater.* 2020, 5, 869-885).

Figure R15 Schematic of RTP performance of monomer and aggregate (*Nat. Rev. Mater.* 2020, 5, 869-885).

(6) **Reviewer #3 wrote:** *The obtained cellulose trimellitate showed relatively good solubility in aqueous solution (Figure 3). Then, if its solids show water/heating-responsive RTP effect for the destructive effect of water to the intermolecular interactions?*

Answer: Thanks for your kind suggestion. We detected the water/heating-responsiveness of CBtCOONa. CBtCOONa exhibited reversible water/heating-responsiveness. The presence of water destroys the hydrogen bonding interactions between adjacent CBtCOONa chains, leading to the destruction of the rigid

environment. Therefore, no RTP emission was detected after water fumigation. When water was removed, the intermolecular hydrogen bonds were constructed again. CBtCOONa exhibited RTP emission again (Figure R16).

Figure R16. Photographs of the reversible heating/water responsiveness process of CBtCOONa.

- (7) **Reviewer #3 wrote:** *Some important references related to this work should be added, such as Sci. Adv. 2022, 8, eabl8392; Chem. Soc. Rev., 2021, 50, 12616; Chinese J. Chem., 2022, 40, 2359; Acc. Chem. Res., 2020, 53, 962 and so on.*

Answer: Thanks for your kind suggestion. We have cited the highly-related work in the revised manuscript.

Answers to Comments by Reviewer # 4

- (1) **Reviewer #4 wrote:** *In this manuscript entitled “Aggregation-Regulated Room-Temperature Phosphorescence Materials with Multi-Mode Emission, Adjustable Excitation-Dependence and Visible-Light Excitation”, You et al. presented RTP materials with multiple emission and special excitation modes by controlling the aggregation state of CBtCOONa. The strategy is simple, novel, and interesting. It provided a new strategy to fabricate RTP materials. The manuscript is well written and describes very thoroughly the synthesis processes and the applications of the RTP materials. It has a wide readership. It is suitable to the scope of NATURE COMMUNICATIONS. I recommend publishing this paper after minor revisions.*

Answer: Thanks for your nice comments and recommendation.

- (2) **Reviewer #4 wrote:** *Are the RTP materials crystal or amorphous? XRD spectra*

of the RTP materials should be shown.

Answer: Thanks for your kind suggestion. XRD spectra of CBtCOOM show that the RTP materials are amorphous (Figure R17a). XRD spectrum of CBtCOONa/CaCO₃ indicates the RTP material has CaCO₃ crystal (Figure R17b).

Figure R17 (a) XRD spectra of CBtCOOM. (b) XRD spectrum of CBtCOONa/CaCO₃ powder with CBtCOONa content of 50%.

(3) **Reviewer #4 wrote:** Except the reported metal ions, what is the effect of other common metal ions, such as Fe²⁺, Fe³⁺, Cu²⁺, Co²⁺ and so on?

Answer: Thanks for your kind suggestion. As shown in Figure R18, the phosphorescence cannot be detected, when the Fe²⁺, Fe³⁺, Cu²⁺, Cu⁺, Co²⁺ and Cr³⁺ ions were used.

Figure R18 Photographs and phosphorescence spectra of CBtCOOM with some common

metal ions.

(4) **Reviewer #4 wrote:** *Cellulose could demonstrate as a chiral polymer. Is the RTP chiral?*

Answer: Thanks for your kind suggestion. We have measured circularly polarized luminescence of CBtCOONa. As shown in Figure R19, there are a strong chiral fluorescence peak at 420 nm and a weak chiral phosphorescence peak at 510 nm, which demonstrate that the CBtCOONa emits chiral RTP.

Figure R19 Circularly polarized luminescence spectra of CBtCOONa.

REVIEWERS' COMMENTS

Reviewer #1 (Remarks to the Author):

The authors have thoroughly addressed all of my concerns, as well as those raised by the other reviewers. The manuscript has been meticulously revised and refined. As such, I highly recommend this version of the manuscript for publication in Nat. Commun., pending the completion of the three minor revisions outlined below.

1. Supplementary Figure 2 should be updated with correct title of abscissa.
2. Please add the measurement temperatures of lifetimes in the caption of Fig. 3c and e.
3. Although the peer review file will be published together, the reasons for the selection of CBtCOOH (DS = 0.54) as the model are suggested to be added in the manuscript.

Reviewer #2 (Remarks to the Author):

It seems that the authors have carefully and fully addressed the concerns of the reviewers. This version is acceptable for publication.

Reviewer #3 (Remarks to the Author):

The revised version is suitable for publication now.

Reviewer #4 (Remarks to the Author):

Authors successfully addressed the reviewers comments. The manuscript can be accepted now.

Answers to Comments by Reviewers

Answers to Comments by Reviewer # 1

- (1) **Reviewer #1 wrote:** *The authors have thoroughly addressed all of my concerns, as well as those raised by the other reviewers. The manuscript has been meticulously revised and refined. As such, I highly recommend this version of the manuscript for publication in Nat. Commun., pending the completion of the three minor revisions outlined below.*

Answer: Thanks for your nice comments and recommendation.

- (2) **Reviewer #1 wrote:** *Supplementary Figure 2 should be updated with correct title of abscissa.*

Answer: Thanks for your kind suggestion. We have revised the abscissa title in the revised Supplementary Information.

- (3) **Reviewer #1 wrote:** *Please add the measurement temperatures of lifetimes in the caption of Fig. 3c and e.*

Answer: Thanks for your kind suggestion. We have added the measurement temperatures in the revised Manuscript.

- (4) **Reviewer #1 wrote:** *Although the peer review file will be published together, the reasons for the selection of CBtCOOH ($DS = 0.54$) as the model are suggested to be added in the manuscript.*

Answer: Thanks for your kind suggestion. We have added the reasons in the revised Manuscript as suggestion.

Answers to Comments by Reviewer # 2

- (1) **Reviewer #2 wrote:** *It seems that the authors have carefully and fully addressed the concerns of the reviewers. This version is acceptable for publication.*

Answer: Thanks for your nice comments and recommendation.

Answers to Comments by Reviewer # 3

- (1) **Reviewer #3 wrote:** *The revised version is suitable for publication now.*

Answer: Thanks for your nice comments and recommendation.

Answers to Comments by Reviewer # 4

- (1) **Reviewer #4 wrote:** *Authors successfully addressed the reviewers comments. The manuscript can be accepted now.*

Answer: Thanks for your nice comments and recommendation.